# Quality Evaluation of Tetrastigmae Radix from Two Different Habitats Based on Simultaneous Determination of Multiple Bioactive Constituents Combined with Multivariate Statistical Analysis

**DOI:** 10.3390/molecules27154813

**Published:** 2022-07-27

**Authors:** Haijie Chen, Yongyi Zhou, Jia Xue, Jiahuan Yuan, Zhichen Cai, Nan Wu, Lisi Zou, Shengxin Yin, Wei Yang, Xunhong Liu, Jianming Cheng, Li Tang

**Affiliations:** 1College of Pharmacy, Nanjing University of Chinese Medicine, Nanjing 210023, China; chenhaijie039x@163.com (H.C.); zyy18851091328@163.com (Y.Z.); 17860505214@163.com (J.X.); yuanjiahuan1027@163.com (J.Y.); caizhichen2008@126.com (Z.C.); wunan7272@163.com (N.W.); yinshengxin723@163.com (S.Y.); yangwei202103@163.com (W.Y.); cjm7895@163.com (J.C.); 13813387461@163.com (L.T.); 2Jiangsu Province Engineering Research Center of Classical Prescription, Nanjing 210023, China

**Keywords:** Tetrastigmae Radix, two different habitats, multiple bioactive constituents, simultaneous determination, multivariate statistical analysis

## Abstract

Tetrastigmae Radix, also known as Sanyeqing (SYQ) in Chinese, is an important traditional Chinese medicine with a long history. *Tetrastigma hemsleyanum* Diels et Gilg mainly grows in the south of the Yangtze River and is widely distributed. The content of bioactive constituents in SYQ varies greatly in different habitats, and there are obvious differences in the content of bioactive constituents between southwestern SYQ (WS) and southeastern SYQ (ES). To distinguish and evaluate the quality of ES and WS, an analytical method based on ultrafast performance liquid chromatography coupled with triple quadrupole-linear ion trap mass spectrometry (UFLC-QTRAP-MS/MS) was established for the simultaneous determination of 60 constituents including 25 flavonoids, 9 phenolic acids, 15 amino acids, and 11 nucleosides in 47 samples from ES and WS. In addition, orthogonal partial least squares discriminant analysis (OPLS-DA), *t*-test, and gray correlation analysis (GRA) were used to discriminate and evaluate the ES and WS samples based on the contents of 60 constituents. The results showed that there were significant differences in the bioactive constituents between ES and WS, and ES was superior to WS in terms of quality evaluation. This study not only provides basic information for differentiating ES and WS but also provides a new perspective for the comprehensive evaluation and quality control of SYQ from two different habitats.

## 1. Introduction

*Tetrastigma hemsleyanum* Diels et Gilg is a green vine of the family Vitaceae, mainly with root parts into medicine named Sanyeqing (SYQ), recorded in Chinese Materia Medica [1]. The Flora of China records that *Tetrastigma hemsleyanum* Diels et Gilg is mainly distributed in Zhejiang, Jiangxi, Fujian, Guangxi, Guizhou, Yunnan, Chongqing, and other provinces south of the Yangtze River in China [2]. Modern phytochemistry shows that SYQ contains a variety of active ingredients, including flavonoids, phenolic acids, nucleosides, amino acids, polysaccharides, triterpenes, steroids, alkaloids, and other constituents [3]. Modern pharmacological research on SYQ has a variety of pharmacological effects, including antitumor, antioxidant, anti-inflammatory, antibacterial, antipyretic, analgesic, immunomodulatory, and other activities [4,5,6,7,8,9,10,11,12,13,14]. Among the total phenolic acids and total flavonoids of SYQ, rutin, isoquercitrin, nicotifiorin, and astragalin are the main substances, for which metabolites in the body could explain some of the antioxidant biomarkers [9]. Amino acids and nucleosides are usually considered to have nutritional value and health benefits [15]. The total amino acids in SYQ have a protective effect against CCl_4_-induced liver injury in mice [16].

The quality of SYQ is affected by factors such as habitat [17], seed source [18], growth environment [19], and processing methods [20], and the appearance of the roots varies significantly, in which the accumulation of chemical constituents is also affected. Few studies have been conducted on these factors. Some studies have found that the habitat factor has a greater influence on the accumulation of constituents in SYQ [17], but they are not comprehensive enough.

At present, many analytical methods have been reported for quality assessment and control in SYQ, such as high-performance liquid chromatography (HPLC) [21,22,23], inductively coupled plasma–mass spectrometry (ICP–MS) [24], and liquid chromatography-mass spectrometry (LC–MS) [4,25,26,27]. More studies have focused on the quantitative analysis of total flavonoids, total phenolic acids, and polysaccharide content, and few have studied the quantitative analysis of phenolic acids, amino acids, and nucleosides of SYQ based on LC–MS. It is necessary to establish a method combining multivariate index constituents to distinguish SYQ from different habitats.

This study aimed to identify and evaluate the quality of southeastern SYQ (ES) and southwestern SYQ (WS) based on the simultaneous determination of multiple bioactive constituents in combination with multivariate statistical analysis. A reliable method based on ultrafast liquid chromatography coupled with triple quadrupole-linear ion trap tandem mass spectrometry (UFLC-QTRAP-MS/MS) for the simultaneous determination of 60 constituents in SYQ was developed. Orthogonal partial least squares discriminant analysis (OPLS-DA) and *t*-test were applied to distinguish and reveal the differential constituents of ES and WS. In addition, gray correlation analysis (GRA) was used to assess the quality of SYQ based on the correlation between the detected component contents and the samples. The established method can provide a basis for a comprehensive evaluation and quality control of SYQ from two different habitats, and it provides fundamental data to distinguish between ES and WS.

## 2. Results

### 2.1. Optimization of Extraction Conditions

Extraction solvent, extraction time, and solid–liquid ratio have important effects on the extraction of target constituents in SYQ. In order to obtain the proper extraction efficiency of rutin, isoquercitrin, nicotifiorin, and astragalin, single-factor tests were performed for extraction time (20, 30, 40, 50, and 60 min), the extraction solvent (50% methanol/ethanol, 60% methanol/ethanol, 70% methanol/ethanol, 80% methanol/ethanol, 90% methanol/ethanol, 100% methanol/ethanol, *v/v*), and solid–liquid ratio (1:10 g/mL, 1:15 g/mL, 1:20 g/mL, 1:25 g/mL, and 1:30 g/mL). Finally, by comparing the extraction yields of the four constituents in ethanol solvent (Appendix A), the best extraction method for UFLC-QTRAP-MS was 0.5 g of sample powder extracted with 60% ethanol (12.5 mL) on an ultrasonic machine for 50 min.

### 2.2. Optimization of UFLC Conditions

To obtain the best chromatographic conditions, UFLC chromatographic conditions such as column, mobile phase, and column temperature were optimized to achieve a higher separation effect and better peak shape of the target constituents in SYQ. The results showed that the separation capacity and sensitivity of the XBridge^®^C18 column (4.6 mm × 100 mm, 3.5 μm) were relatively superior. In addition, five mobile phase systems (water–methanol, water–acetonitrile, water–methanol:acetonitrile (1:1), 0.1%, 0.4%, 0.8% formic acid water solution–methanol solution, and 0.1%, 0.4%, 0.8% formic acid water solution–acetonitrile solution), flow rates (0.3, 0.4, 0.5, 0.7, 0.8, 0.9, 1.0 mL/min), and column temperatures (25, 30, 35, 40 °C) were examined and compared. The expected separation was achieved by gradient elution with 0.4% formic acid as Eluent A and methanol as Eluent B at a flow rate of 0.8 mL/min under the column temperature of 30 °C.

### 2.3. Optimization of Mass Spectrometry (MS) Conditions

The individual solutions of all standard compounds (about 100 ng/mL) were examined with the electrospray ionization (ESI) source in the positive and negative ion modes. After repeated experimental tests, amino acids and nucleosides showed good sensitivity and intensity in the positive ion mode, while flavonoids and phenolic acids were more suitable for detection in the negative ion mode. Uridine responded better in negative ion mode than in positive ion, and Orientin and Iso-orientin responded better in positive ion mode than in negative ion mode [28]. Therefore, both ESI+ and ESI− modes were used in this study. Although the retention times of some constituents were similar, they could be precisely quantified based on different precursor and product ion pairs. Table 1 lists the best details of the 60 constituents in terms of retention time (t_R_), precursor and product ions, declustering potential (DP), and collision energy (CE). The MS spectra of 32 constituents in negative ion mode are shown in Appendix A, and the MS spectra of 28 constituents in positive ion mode are shown in Appendix A. Figure 1 shows the multiple reaction monitoring (MRM) for the 60 constituents.

### 2.4. Method Validation

All method validations of quantification were performed by the established UFLC-QTRAP-MS/MS method. The detailed results of each method validation are presented in Table 2. Each standard calibration curve was constructed by plotting the peak areas (*Y*) against the corresponding concentrations (*X*). All analytes showed good linearity with appropriate determination coefficients (r > 0.9989). The ranges of limits of detection and quantification (LODs and LOQs) were 0.03–13.59 ng/mL and 0.09–45.3 ng/mL, respectively. The relative standard deviations (RSDs) of intraday and interday variations ranged from 0.93% to 4.97% and 0.88% to 4.97%, respectively. The RSDs of the repeatability and stability were less than 4.98% and 4.99%, respectively. The overall recoveries varied from 96.1% to 101.76%, with RSDs < 4.87%. The slope ratio values of the matrix curve to the pure solution curve were between 0.92 and 1.05, indicating that the matrix effect on the ionization of analytes was not obvious under optimized conditions.

### 2.5. Quantitative Analysis of Samples

Sample information is shown in Figure 2. The validated analytical method of UFLC-QTRAP-MS/MS was successfully applied to simultaneously determine 60 constituents (25 flavonoids, 9 phenolic acids, 15 amino acids, and 11 nucleosides) in SYQ. The quantitative results of 60 constituents are presented in Appendix A. The SYQ samples were all rich in amino acids, with total amino acid contents ranging from 360.04 to 2856.77 µg/g, accounting for more than 65% of the total analyte content in this study. In addition, the contents of Proline (**10**), Alanine (**5**), Phenylalanine (**27**), and Lysine (**1**) were relatively high. The total content of nucleosides ranged from 40.05 to 246.80 µg/g, with Adenosine (**16**), Hypoxanthine (**14**), Uridine (**15**), and Uracil (**12**) accounting for more than 86% of the total nucleoside content. The total content of phenolic acids was 4.33–134.78 µg/g, of which the content of Piceatannol (**42**) accounted for more than 58%. The total content of flavonoids was 26.2–2361.67 µg/g, of which Procyanidin B2 (**30**), Catechin (**33**), Nicotifiorin (**53**), Rutin (**48**), Isoquercitrin (**49**), Astragalin (**52**), Epicatechin (**38**) were relatively high, and **30** accounted for more than 64% of the total flavonoids. The content of **30**, **33**, and **38** in the ES sample was higher than that in the WS sample, while **52**, **48**, and **49** were on the contrary. Figure 3 shows that the total contents of amino acids, nucleosides, phenolic acids, and flavonoids in ES were significantly higher than those in WS.

### 2.6. OPLS-DA of Samples

Firstly, principal component analysis (PCA) was used to differentiate and assess the quality of ES and WS. Since Principal Component Analysis (PCA) could not clearly reflect the classification of the measured samples, it was not suitable to provide a basis for differentiating and evaluating the quality of ES and WS. Orthogonal partial least squares discriminant analysis (OPLS-DA) is a supervised latent structure discriminant analysis method that maximizes between-group variation and minimizes within-group separation. Orthogonal partial least squares discriminant analysis (OPLS-DA) is a supervised latent structure discriminant analysis method that maximizes between-group variation and minimizes within-group separation. The method maximizes group differences and minimizes within-group separation. Figure 4 shows the OPLS-DA score plot. ES and WS were divided into two groups, thus indicating significant differences in chemical composition between them. R2 describes the degree of fitting of the model. Q2 describes X’s ability to predict Y. It is generally believed that Q2 greater than 0.5 indicates that the model has good reliability and predictability, and greater than 0.9 is excellent [29]. In this comparison, the statistical parameters of OPLS-DA R2X(cum), R2Y(cum), and Q2(cum) are 0.767, 0.931, and 0.895, respectively, indicating that the model has good repeatability and predictability. The variable importance of projection (VIP) is a vector summarizing the total importance of variables in explaining the model. If a variable’s VIP > 1, it indicates that the variable contributes significantly to the classification of these samples. As shown in Figure 5, according to the VIP value, eight constituents were found to play a dominant role in the cluster, including Lysine (**1**), Histidine (**2**), Alanine (**5**), Proline (**10**), Leucine (**25**), Phenylalanine (**27**), Procyanidin B2 (**30**), and Catechin (**33**).

### 2.7. T-Test of Samples

*T*-test was used to analyze the contents of bioactive constituents detected to evaluate the changes of 60 constituents in ES and WS, and it was considered that the values with *p* values less than 0.05 had significant differences. As shown in Figure 6, more than half of the bioactive constituents in ES were higher (*p* < 0.05) than those in WS. The contents of Lysine, Histidine, Glycine, Serine, Alanine, Aspartic acid, Threonine, Cytidine, Guanosine, Isoleucine, Leucine, Phenylalanine, Procyanidin B2, Catechin, Procyanidin B1, Epicatechin, and Polydatin were significantly higher (*p* < 0.001) in ES than that in WS. However, the content of 2′-Deoxyadenosine was higher (*p* < 0.001) in WS than that in ES. In combination with OPLS-DA analysis, Procyanidin B2, Catechin, Lysine, Histidine, Alanine, Leucine, and Phenylalanine could be the most effective chemical markers to distinguish ES and WS.

### 2.8. GRA of Samples

GRA is a measure of influence in gray systems theory that analyzes the uncertain relationship between a major factor and all other factors in a given system. Therefore, a comprehensive GRA evaluation of ES and WS was performed based on the content of 60 bioactive constituents. The GRA results including the gray comprehensive evaluation value (*r_i_*) and the quality rankings are shown in Table 3. The *r_i_* indicates the relative correlation between component content and samples. The samples with higher relative correlation are of better quality.

Table 3 shows the grey comprehensive evaluation value (*r_i_*) and the quality ranking. From this perspective, the overall quality of ES was significantly better than that of WS. SYQ produced in Zhejiang was of better quality relative to other provinces. In addition, among the southwestern habitat, SYQ produced in Guangxi was better and those produced in Yunnan and Guizhou were worse. The difference of *r_i_* varied widely, with a maximum value of 38.67%, which could well-distinguish the quality of the samples. However, there were differences in the quality of SYQ from different provinces, which could be attributed to latitude, altitude, and harvest time. In summary, as can be seen in Table 3, GRA can successfully assess the quality of SYQ based on the content of its multiple components.

## 3. Discussion

A highly efficient and reliable UFLC-QTRAP-MS/MS method was developed for the simultaneous determination of 25 flavonoids, 9 phenolic acids, 15 amino acids, and 11 nucleosides in 47 samples. The OPLS-DA, VIP values, and *t*-test indicated that there were significant differences in the bioactive constituents in ES and WS, such as procyanidin B2 (**30**), catechin (**33**), lysine (**1**), histidine (**2**), alanine (**5**), leucine (**25**), and phenylalanine (**27**), which can be considered to distinguish and control the quality of ES and WS (Figure 5 and Figure 6). Compared with existing research methods [21,22,23,24], this method analyzes a large number of active constituents, including pharmacological and nutritional constituents. At the same time, the mass spectrometry detector uses the positive and negative ion mode for simultaneous determination, which allows precise determination of the molecular weight of the constituents and is more sensitive than conventional detectors such as UV detectors. This method can overcome the shortcomings of traditional detection methods and effectively reveal the complexity of sample composition, but it also has some limitations. Since the UFLC-QTRAP-MS/MS method is suitable for the determination of small molecules and constituents with low boiling points, there are difficulties in the identification of constituents with large molecular weights and volatile constituents. Though this drawback exists, the 60 constituents with small molecular weight and high boiling point detected in this experiment showed good response values in this method.

Amino acids are nutritional constituents, and the total amino acid content is higher than 50% in SYQ (Figure 3). As the total amino acids in SYQ have some hepatic protective effects and are poorly studied [16], the UFLC-QTRAP-MS/MS method established in this study provides a basis for the amino acid content determination. From Figure 5, the VIP-value of constituents **2**, **10**, and **25** is close, with a range of 1.0–1.5. Meanwhile, for constituents **1**, **5**, **27**, **30**, and **33,** VIP is > 2. Procyanidin B2 (**30**) and catechin (**33**) are flavonoids. The flavonoid constituents of SYQ have antitumor [5] and antioxidant [7] effects, and these constituents may provide a new idea for the study of the antitumor activity of SYQ from different origins. Based on the *r_i_* values of the samples (Table 3), the overall quality of ES was better than WS, indicating that different geographical regions can influence the accumulation of bioactive constituents. Meanwhile, the GRA data showed some differences in samples from the same habitat, possibly related to factors such as geographic environment and cultivation techniques. In conclusion, the UFLC-QTRAP-MS/MS method combined with multivariate statistical analysis can provide basic information for the identification and quality evaluation of SYQ from different habitats.

## 4. Materials and Methods

### 4.1. Plant Materials

Plant materials were collected from nine provinces, including Zhejiang, Fujian, Jiangxi, Hunan, Hubei, Guangxi, Guizhou, Yunnan, and Chongqing. Table 4 shows the detailed geographic habitats of each sample. All the samples were authenticated by Professor Xunhong Liu (Nanjing University of Chinese Medicine, Nanjing, China) and were deposited in the laboratory of Chinese medicine identification, Nanjing University of Chinese Medicine. Detailed information is shown in Table 4.

### 4.2. Chemicals and Reagents

The standards of Uridine (**15**), Epicatechin (**38**), Quercitrin (**51**), Quercetin (**56**), Gallic acid (**21**), and Kaempferol (**58**) were purchased from the Chinese National Institute of Control of Pharmaceutical and Biological Products (Beijing, China). Phenylalanine (**27**), Glutamic acid (**8**), Hyperoside (**46**), Caffeic acid (**37**), Leucine (**25**), Vitexin (**43**), Guanosine (**19**), Proline (**10**), Threonine (**7**), Serine (**4**), Adenosine (**16**), Valine (**13**), Isoleucine (**23**), and Isorhamnetin (**60**) were purchased from the Institute of Food and Drug Administration of China (Beijing, China). 2′-Deoxyadenosine (**17**), 2′-Deoxyinosine (**24**), 2′-Deoxyguanosine (**22**), Resveratrol (**50**), Cytidine (**11**), Alanine (**5**), Hypoxanthine (**14**), Catechin (**33**), Glycine (**3**), Polydatin (**39**), Inosine (**20**), Tyrosine (**18**), Luteolin (**57**), Uracil (**12**), Apigenin (**59**), Aspartic acid (**6**), Thymidine (**26**), Protocatechuic acid (**28**), and Histidine (**2**) were purchased from Yuanye Biotechnology Co., Ltd. (Shanghai, China). 3,4-Dihydroxybenzoic acid (**31**) was purchased from Shanghai Ronghe Pharmaceutical Technology Co., Ltd. (Shanghai, China). Rutin (**48**) was purchased from the China National Institute for the Control of Biological Products (Beijing, China). Isoquercitrin (**49**) was purchased from Jiangsu Yongjian Pharmaceutical Technology Co., Ltd. (Jiangsu, China). Epigallocatechin (**32**), Orientin (**40**), and Isoorientin (**41**) were purchased from Chengdu Herb Purify Biotechnology Co., Ltd. (Chengdu, China). Procyanidin B2 (**30**), Procyanidin B1 (**34**), Isovitexin (**45**), Astragalin (**52**), and Nicotifiorin (**53**) were purchased from Chengdu Desite Biotechnology Co., Ltd. (Chengdu, China). Neochlorogenic acid (**29**) and Cryptochlorogenic acid (**36**) were purchased from Chengdu Purifa Technology Development Co., Ltd. (Chengdu, China). Piceatannol (**42**), Vitexin-2″-O-rhamnoside (**44**), Afzelin (**55**), Aromadendrin (**47**), and Narcissin (**54**) were purchased from Chengdu Alfa Biotechnology Co., Ltd. (Chengdu, China). Lysine (**1**), Cysteine (**9**), and Chlorogenic acid (**35**) were purchased from Baoji Chenguang Biotechnology Co., Ltd. (Baoji, China).

### 4.3. Preparation of Standard Solutions

A mixed standard stock solution containing 60 reference standards was prepared with 70% ethanol and with the following concentrations: 1.008 (**1**), 1.000 (**2**), 1.160 (**3**), 1.180 (**4**), 1.100 (**5**), 1.180 (**6**), 1.160 (**7**), 0.960 (**8**), 1.024 (**9**), 1.046 (**10**), 0.962 (**11**), 1.010 (**12**), 1.010 (**13**), 0.998 (**14**), 1.064 (**15**), 0.784 (**16**), 0.970 (**17**), 1.000 (**18**), 0.990 (**19**), 1.104 (**20**), 0.970 (**21**), 1.016 (**22**), 1.070 (**23**), 1.078 (**24**), 1.012 (**25**), 1.014 (**26**), 1.164 (**27**), 1.036 (**28**), 0.960 (**29**), 1.096 (**30**), 1.154 (**31**), 1.034 (**32**), 1.044 (**33**), 1.075 (**34**), 0.984 (**35**), 0.662 (**36**), 1.052 (**37**), 1.034 (**38**), 1.062 (**39**), 1.034 (**40**), 1.168 (**41**), 0.996 (**42**), 1.118 (**43**), 1.056 (**44**), 0.994 (**45**), 0.936 (**46**), 0.970 (**47**), 1.020 (**48**), 1.094 (**49**), 1.036 (**50**), 1.032 (**51**), 1.004 (**52**), 0.992 (**53**), 0.982 (**54**), 0.866 (**55**), 1.032 (**56**), 1.128 (**57**), 0.320 (**58**), 1.064 (**59**), 1.062 (**60**) mg/mL; then, they were diluted with 70% ethanol to different concentrations to generate the calibration curves. All of the solutions were stored at 4 °C before LC–MS analysis.

### 4.4. Preparation of Sample Solutions

The sample powder was weighed precisely at about 0.5 g; then, it was sonicated with 60% ethanol (12.5 mL) for 50 min, cooled, made up for the weight loss with 60% ethanol, shaken well, and filtered; the filtrate was centrifuged at 12,000 r/min for 10 min, and the supernatant was filtered through a 0.22 μm microporous membrane and stored at 4 °C in a refrigerator.

### 4.5. Chromatographic and Mass Spectrometric Conditions

The chromatographic analysis of SYQ was performed on a SIL-20A XR system (Shimadzu, Kyoto, Japan). The separation was conducted by a XBridge^®^C18 column (4.6 mm × 100 mm, 3.5 μm) at 30 °C and the injection volume was 2 µL. The mobile phase contained 0.4% formic acid water solution (A) and methanol solution (B) at 0.8 mL/min flow rate with the following gradient elutions: 0–4 min, 7–9% B; 4–6 min, 9–21% B; 6–10 min, 21–35% B; 10–12 min, 35–38% B; 12–16 min, 38–46% B; 16–20 min, 46–64% B; 20–21 min, 64–7% B. 

An API5500 triple quadrupole linear ion trap tandem mass spectrometer (AB SCIEX, Framingham, MA, USA) equipped with an electrospray ionization (ESI) source was used for detection. The operating parameters were as follows: ion source temperature, 550 °C; nebulizer gas (GS1) flow, 55 L/min; auxiliary gas (GS2) flow, 55 L/min; curtain gas (CUR) flow, 40 L/min; spray voltage (IS), 4500 V in the positive mode and −4500 V in the negative mode. Detection of analytes was performed in multiple-reaction mode (MRM).

### 4.6. Validation of the Method

The method was validated in terms of linearity, precision of intraday and interday, repeatability, stability, recovery, and matrix effect. Serial dilutions of mixed standards were used to establish the standard curves, and the linear regression equation, correlation coefficient, and linear range were calculated. The detection limit (LOD) and quantification limit (LOQ) for 33 constituents were calculated at the signal-to-noise ratios of 3 and 10, respectively. For intraday precision, the mixed standards solutions were injected for six replicates within one day, while for interday precision, the solutions were examined in triplicates for 3 consecutive days. To validate the repeatability, six samples of SYQ were accurately weighed and prepared independently according to the optimal conditions above and then analyzed. The same sample solution was taken and determined at 0, 2, 4, 8, 12, and 24 h according to the above chromatographic conditions to evaluate the stability. The recovery experiments were used to assess the accuracy of the method; standards at three different concentration levels, including low (80%), median (100%), and high (120%) were added to samples of known content. Each experiment was repeated three times, and the spiked samples were analyzed by UFLC-QTRAP-MS/MS to evaluate the recoveries. The recoveries were calculated by the formulae: recovery (%) = (detected amount − original amount)/spiked amount × 100%. The matrix effect refers to the enhancement or suppression of a chromatography signal by interference or coeluting constituents in the matrix. It was evaluated using a slope comparison method. In this way, the matrix effect was determined to be the ratio of the slope in a matrix-matched calibration curve to the slope in a solvent standard curve. The slope ratio close to 1.0 indicates that the matrix effect is weaker.

### 4.7. Multivariate Statistical Analysis

After data preprocessing, the global clustering trend of each group was observed by applying OPLS-DA and its distribution was visualized using SIMCA-P 13.0 software (Umetrics AB, Umea, Sweden). Further, statistical analysis of all detected component data was performed by *t*-test (SPSS 16.0 for Windows, IBM, Armonk, NY, USA) to detect differential constituents between ES and WS. Based on the results of quantitative measurements and *t*-tests, box line plots were created by OriginPro 2021b (OriginLab, Northampton, MA, USA) to obtain metabolite distribution maps and to analyze the differences between ES and WS. The quality of ES and WS samples was assessed based on the content of 60 active constituents using GRA, using Excel for Mac 2019 (Microsoft Corporation, Seattle, WA, USA). OriginPro 2021b (OriginLab, Northampton, MA, USA) was used to plot all histograms.

## 5. Conclusions

A reliable UFLC-QTRAP-MS/MS method was developed for the simultaneous determination of 60 constituents including flavonoids, phenolic acids, amino acids, and nucleosides in SYQ. Furthermore, multivariate statistical analyses such as OPLS-DA analysis, *t*-test, and GRA were applied to comprehensively analyze and evaluate different habitats of SYQ (ES and WS). The OPLS-DA analysis and *t*-test were applied to classify and identify SYQ from different habitats. It was found that ES and WS differed significantly and their classification was related to the differential constituents, such as procyanidin B2 (**30**), catechin (**33**), lysine (**1**), histidine (**2**), alanine (**5**), leucine (**25**), and phenylalanine (**27**), which could be used as chemical markers to distinguish ES between WS. In addition, the GRA results showed that ES was better in quality based on the content of 60 constituents. These results suggest that the accumulation and quality of bioactive constituents in SYQ are influenced by different habitats. This research not only provides a foundation for distinguishing ES and WS but also for the comprehensive evaluation and quality control of SYQ in different habitats.

## Figures and Tables

**Figure 1 molecules-27-04813-f001:**
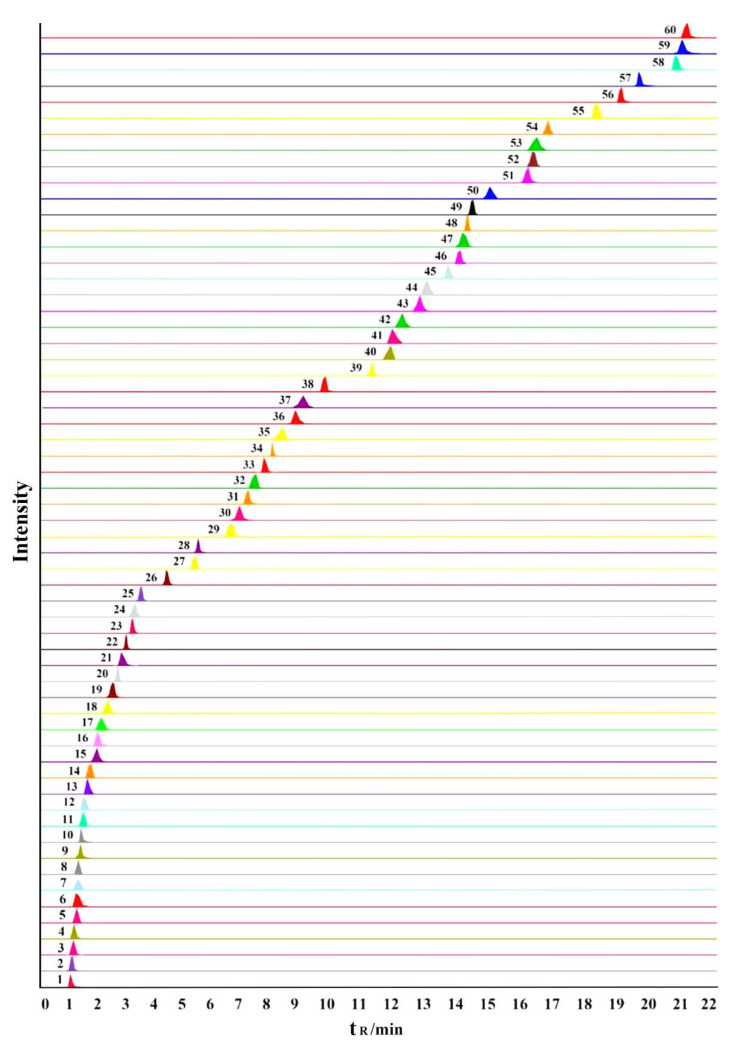
Representative extract ion chromatograms (XIC) of multiple-reaction monitoring (MRM) chromatograms of the 60 investigated constituents. (The peak numbers denoted are the same as those in Table 1.).

**Figure 2 molecules-27-04813-f002:**
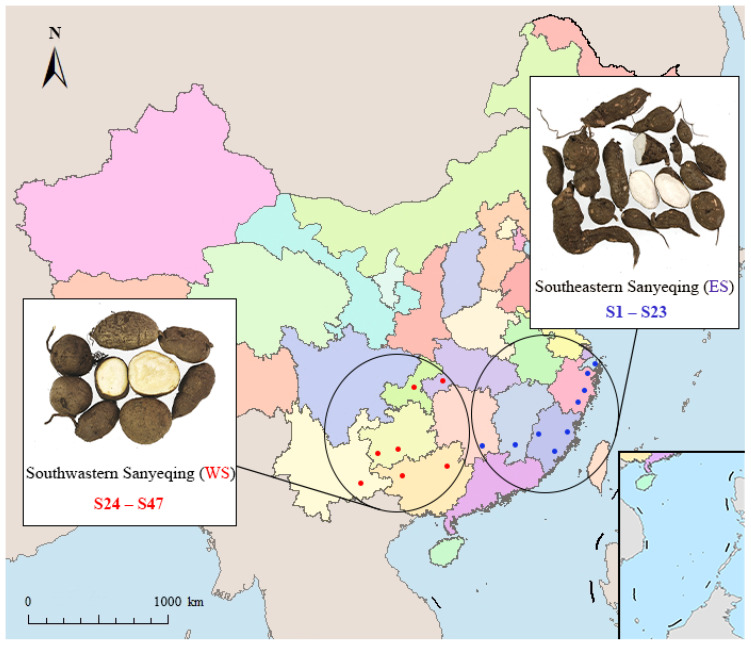
SYQ of two different habitats (ES, WS). The ES samples (S1–S23) were from nine origins and the WS samples (S24–S47) were from seven origins. The origin information of the samples are presented in Table 4.

**Figure 3 molecules-27-04813-f003:**
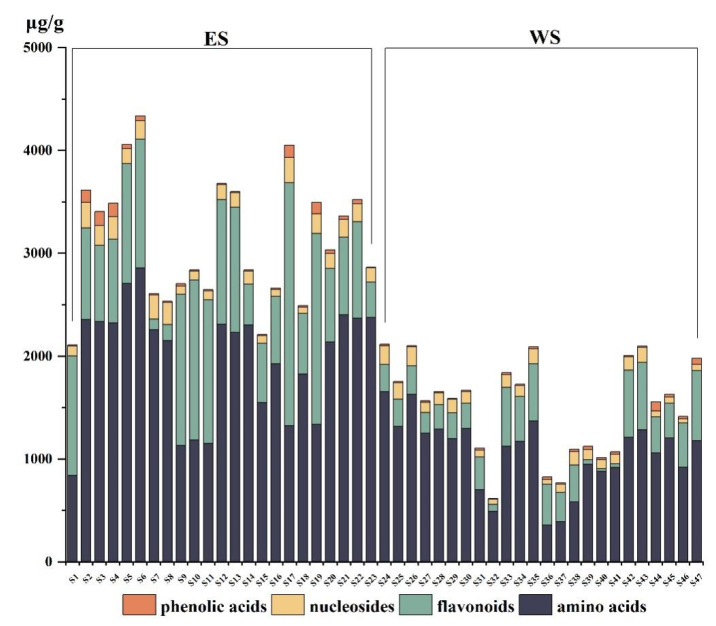
The contents of four kinds of chemical constituents in ES and WS.

**Figure 4 molecules-27-04813-f004:**
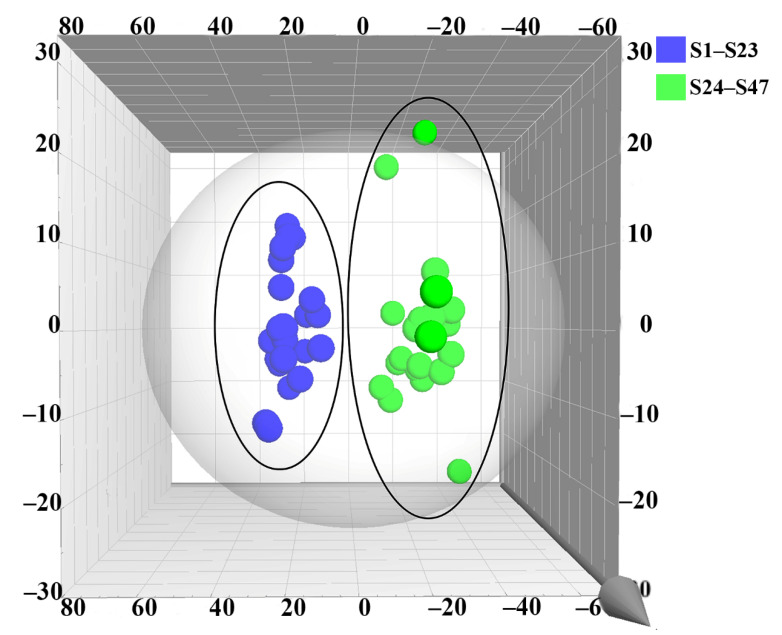
The OPLS-DA model for the classification of ES and WS is based on the content of 60 constituents.

**Figure 5 molecules-27-04813-f005:**
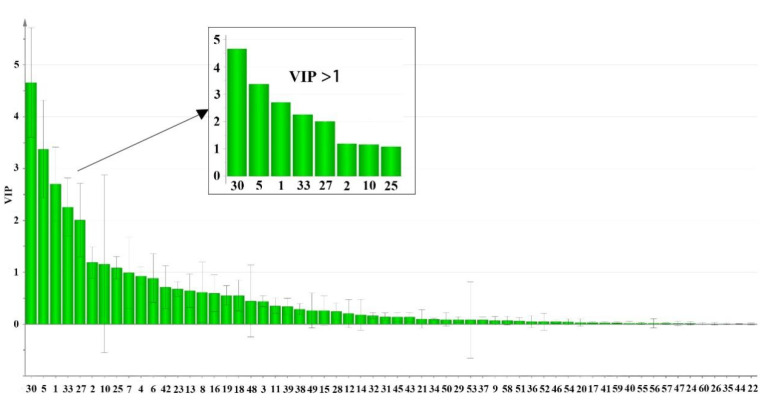
VIP for classification of ES and WS. The *X*-axis numbers denoted are the same as those in Table 1. The eight constituents of VIP >1 are shown in the enlarged graph.

**Figure 6 molecules-27-04813-f006:**
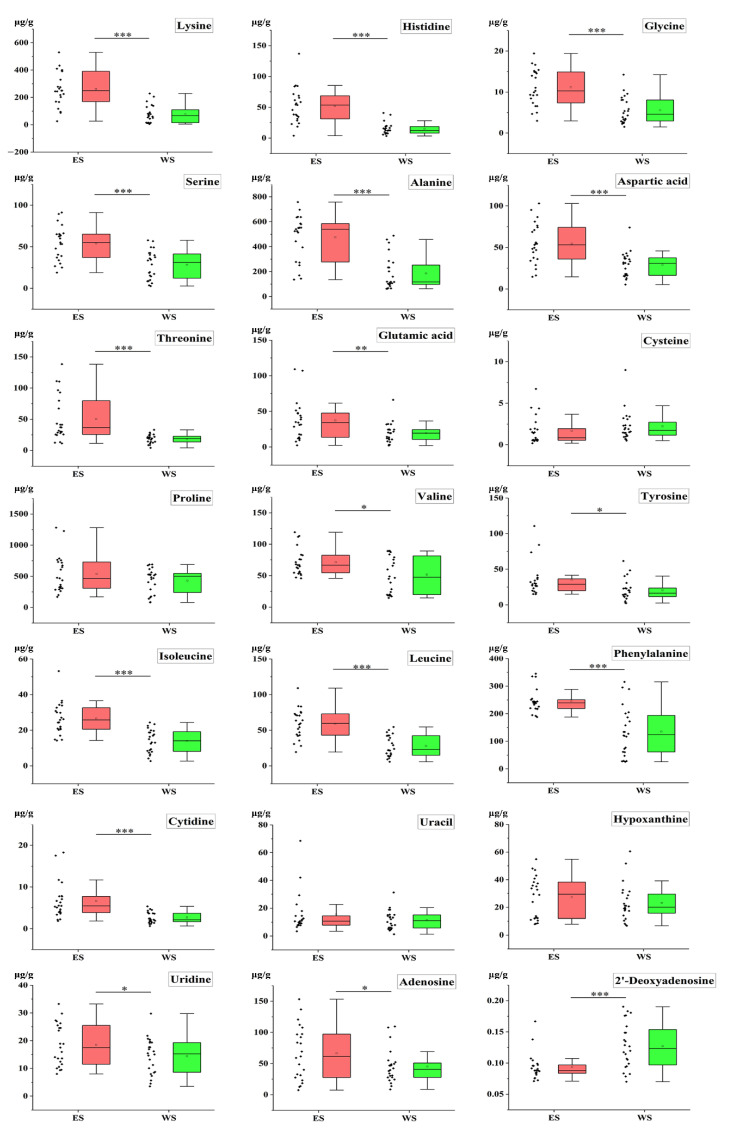
The box plot of 60 constituents’ contents in ES and WS (* *p* < 0.05, ** *p* < 0.01, *** *p* < 0.001).

**Table 1 molecules-27-04813-t001:** Optimized mass spectrometric parameters for MRM of 60 constituents.

No.	Constituents	Formula	t_R_ (min)	Precursor Ion (*m*/*z*)	Product Ion (*m*/*z*)	DP (V)	CE (eV)	CXP (eV)	Ion Mode
1	Lysine	C_6_H_14_N_2_O_2_	1.14	147.11	83.91	100	14	14	ESI+
2	Histidine	C_6_H_9_N_3_O_2_	1.14	156.08	110.03	100	16	14	ESI+
3	Glycine	C_2_H_5_NO_2_	1.20	76.04	30.00	73	6	14	ESI+
4	Serine	C_3_H_7_NO_3_	1.20	106.05	59.99	100	8	14	ESI+
5	Alanine	C_3_H_7_NO_2_	1.26	90.06	44.02	100	10	14	ESI+
6	Aspartic acid	C_4_H_7_NO_4_	1.26	134.05	87.96	59	10	14	ESI+
7	Threonine	C_4_H_9_NO_3_	1.26	120.17	74.00	100	20	14	ESI+
8	Glutamic acid	C_5_H_9_NO_4_	1.26	148.10	83.90	12	14	14	ESI+
9	Cysteine	C_3_H_7_NO_2_S	1.32	122.03	75.93	85	17	14	ESI+
10	Proline	C_5_H_9_NO_2_	1.32	116.07	70.02	68	10	14	ESI+
11	Cytidine	C_9_H_13_N_3_O_5_	1.39	244.09	112.00	61	10	14	ESI+
12	Uracil	C_4_H_4_N_2_O_2_	1.63	113.00	70.00	111	21	14	ESI+
13	Valine	C_5_H_11_NO_2_	1.75	118.09	72.06	100	10	14	ESI+
14	Hypoxanthine	C_5_H_4_N_4_O	1.76	137.00	119.03	80	27	12	ESI+
15	Uridine	C_9_H_12_N_2_O_6_	2.04	243.01	199.96	−115	−14	−13	ESI−
16	Adenosine	C_10_H_13_N_5_O_4_	2.05	268.10	136.10	31	23	14	ESI+
17	2′-Deoxyadenosine	C_10_H_13_N_5_O_3_	2.42	251.81	136.08	80	9	6	ESI+
18	Tyrosine	C_9_H_11_NO_3_	2.66	182.10	136.00	16	16	14	ESI+
19	Guanosine	C_10_H_13_N_5_O_5_	2.77	284.30	152.10	42	16	14	ESI+
20	Inosine	C_10_H_12_N_4_O_5_	2.89	269.00	137.07	46	15	14	ESI+
21	Gallic acid	C_7_H_6_O_5_	2.94	169.00	125.00	−35	−15	−15	ESI−
22	2′-Deoxyguanosine	C_10_H_13_N_5_O_4_	3.02	268.10	152.10	61	15	14	ESI+
23	Isoleucine	C_6_H_13_NO_2_	3.24	132.10	86.05	64	10	14	ESI+
24	2′-Deoxyinosine	C_10_H_12_N_4_O_4_	3.25	253.02	136.90	11	11	16	ESI+
25	Leucine	C_6_H_13_NO_2_	3.58	132.10	86.05	100	16	14	ESI+
26	Thymidine	C_10_H_14_N_2_O_5_	4.72	243.10	127.07	61	13	14	ESI+
27	Phenylalanine	C_9_H_11_NO_2_	5.75	166.10	120.05	100	14	14	ESI+
28	Protocatechuic acid	C_7_H_6_O_4_	5.88	152.90	109.00	−85	−18	−15	ESI−
29	Neochlorogenic acid	C_16_H_18_O_9_	7.03	353.02	190.96	−90	−24	−21	ESI−
30	Procyanidin B2	C_30_H_26_O_12_	7.32	579.20	291.10	120	13	14	ESI+
31	3,4-Dihydroxybenzaldehyde	C_7_H_6_O_3_	7.71	137.00	108.00	−53	−30	−15	ESI−
32	Epigallocatechin	C_15_H_14_O_7_	7.97	305.08	125.02	−155	−26	−55	ESI−
33	Catechin	C_15_H_14_O_6_	8.21	289.00	244.80	−135	−20	−15	ESI−
34	Procyanidin B1	C_30_H_26_O_12_	8.69	577.07	288.96	−185	−32	−15	ESI−
35	Chlorogenic acid	C_16_H_18_O_9_	8.95	353.05	191.03	−120	−22	−13	ESI−
36	Cryptochlorogenic acid	C_16_H_18_O_9_	9.44	353.07	191.01	−105	−20	−21	ESI−
37	Caffeic acid	C_9_H_8_O_4_	9.75	179.03	134.60	−125	−20	−15	ESI−
38	Epicatechin	C_15_H_14_O_6_	10.12	289.00	244.80	−135	−20	−15	ESI−
39	Polydatin	C_20_H_22_O_8_	11.94	389.00	226.90	−140	−27	−15	ESI−
40	Orientin	C_21_H_20_O_11_	12.29	449.20	329.20	35	25	14	ESI+
41	Isoorientin	C_21_H_20_O_11_	12.30	449.20	299.10	35	25	14	ESI+
42	Piceatannol	C_14_H_12_O_4_	12.69	243.00	159.03	−200	−34	−15	ESI−
43	Vitexin	C_21_H_20_O_10_	13.17	431.10	310.90	−100	−30	−15	ESI−
44	Vitexin-2″-O-rhamnoside	C_27_H_30_O_14_	13.36	577.00	413.00	−100	−30	−15	ESI−
45	Isovitexin	C_21_H_20_O_10_	14.03	431.10	311.05	−80	−35	−15	ESI−
46	Hyperoside	C_21_H_20_O_12_	14.46	463.00	300.00	−160	−36	−15	ESI−
47	Aromadendrin	C_15_H_12_O_6_	14.59	286.66	125.02	−165	−26	−13	ESI−
48	Rutin	C_27_H_30_O_16_	14.69	609.06	300.00	−170	−48	−15	ESI−
49	Isoquercitrin	C_21_H_20_O_12_	14.76	463.00	300.00	−180	−36	−15	ESI−
50	Resveratrol	C_14_H_12_O_3_	15.27	227.00	142.70	−150	−35	−15	ESI−
51	Quercitrin	C_21_H_20_O_11_	16.61	447.00	301.00	−165	−30	−15	ESI−
52	Astragalin	C_21_H_20_O_11_	16.84	447.10	283.90	−100	−36	−15	ESI−
53	Nicotifiorin	C_27_H_30_O_15_	16.86	593.19	285.03	−300	−40	−19	ESI−
54	Narcissin	C_28_H_32_O_16_	17.18	623.00	315.00	−240	−30	−15	ESI−
55	Afzelin	C_21_H_20_O_10_	18.79	431.10	285.00	−130	−40	−13	ESI−
56	Quercetin	C_15_H_10_O_7_	19.28	301.10	151.00	−62	−28	−15	ESI−
57	Luteolin	C_15_H_10_O_6_	19.93	285.09	132.98	−170	−40	−15	ESI−
58	Kaempferol	C_15_H_10_O_6_	21.01	285.00	116.90	−120	−36	−15	ESI−
59	Apigenin	C_15_H_10_O_5_	21.21	268.80	116.90	−129	−40	−15	ESI−
60	Isorhamnetin	C_16_H_12_O_7_	21.27	315.00	300.00	−150	−20	−15	ESI−

**Table 2 molecules-27-04813-t002:** Regression equations, limits of detection (LOD), limits of quantification (LOQ), precision, repeatability, stability, recovery, and matrix effect of 60 constituents.

No.	Constituents	Regression Equation	r	Liner Range (ng/mL)	LOD (ng/mL)	LOQ (ng/mL)	Precision (RSD, %)		Repeatability (RSD, %) (*n* = 6)	Stability (RSD, %) (*n* = 6)	Recovery (%)		Matrix Effect
							Intra-Day (*n* = 6)	Inter-Day (*n* = 9)			Mean	RSD	
1	Lysine	Y = 690X + 33,500	0.9990	52.90–20,160	7.96	26.53	4.69	4.80	2.74	1.31	99.79	1.11	0.96
2	Histidine	Y = 2650X − 108,000	0.9990	43.10–6000	0.96	3.19	4.73	4.74	4.58	2.09	99.94	1.41	1.01
3	Glycine	Y = 94.9X + 2210	0.9992	3.05–5800	0.88	2.93	4.54	4.90	2.51	4.69	96.10	4.27	0.97
4	Serine	Y = 439X + 18,700	0.9998	2.89–35,400	0.09	0.31	4.96	4.64	3.73	3.17	96.21	2.08	0.99
5	Alanine	Y = 300X + 1700	0.9998	9.26–22,000	2.73	9.10	4.43	4.21	4.96	4.93	99.77	0.60	0.98
6	Aspartic acid	Y = 374X − 84.9	0.9993	9.48–9440	2.43	8.10	4.91	4.83	4.42	2.48	97.46	2.55	1.02
7	Threonine	Y = 245X + 6090	0.9995	44–5800	10.50	35.00	4.11	4.44	4.84	4.41	97.85	2.09	0.99
8	Glutamic acid	Y = 1490X + 17,400	0.9990	5.59–4800	1.53	5.10	4.84	4.66	4.59	3.80	99.05	1.63	1.02
9	Cysteine	Y = 106X − 52.3	0.9995	22.40–5120	2.13	7.10	3.75	3.07	4.54	4.99	98.43	1.40	0.92
10	Proline	Y = 591X + 28,800	0.9991	13.40–31,380	0.38	1.26	4.70	4.21	4.98	4.85	99.66	1.67	1.03
11	Cytidine	Y = 3530X + 9460	0.9991	0.42–4810	0.12	0.40	4.85	4.97	4.88	1.87	101.76	4.75	0.93
12	Uracil	Y = 284X − 2720	0.9992	18.70–30,300	3.18	10.60	3.46	3.54	4.59	2.40	101.28	4.73	1.05
13	Valine	Y = 1650X − 29,700	0.9992	21.90–10,100	5.91	19.70	1.87	2.03	2.78	4.84	98.56	1.28	1.02
14	Hypoxanthine	Y = 269X + 482	0.9996	18.40–4990	4.98	16.60	4.91	4.77	3.92	2.34	98.71	1.14	0.97
15	Uridine	Y = 301X − 13,500	0.9991	46–21,280	4.04	13.47	0.93	1.00	4.72	3.54	97.90	1.88	0.95
16	Adenosine	Y = 4780X + 5990	0.9995	6.28–15,680	0.93	3.10	2.91	2.74	2.54	4.22	97.75	2.01	1.04
17	2′-Deoxyadenosine	Y = 3810X + 9950	0.9998	4.21–4850	0.45	1.50	4.41	4.78	3.64	4.31	97.57	2.19	1.05
18	Tyrosine	Y = 678X − 6340	0.9999	16.30–50,000	0.16	0.54	3.12	2.65	3.16	2.06	96.77	2.91	0.99
19	Guanosine	Y = 4540X − 11,000	0.9999	2.95–4950	0.39	1.30	3.25	2.79	4.82	4.65	98.21	2.78	0.93
20	Inosine	Y = 3020X − 12,900	0.9999	6.57–5520	1.82	6.07	3.02	2.84	3.64	4.19	98.70	2.26	0.95
21	Gallic acid	Y = 1610X − 10,300	0.9993	11.10–4850	3.01	10.03	1.77	1.51	4.92	4.86	98.36	2.46	1.02
22	2′-Deoxyguanosine	Y = 5990X − 11,300	0.9994	3.08–5080	0.31	1.02	2.71	2.29	1.90	4.67	98.10	1.69	1.01
23	Isoleucine	Y = 4210X + 3820	0.9998	1.94–535	0.56	1.86	2.80	2.67	1.82	2.51	96.12	1.33	0.94
24	2′-Deoxyinosine	Y = 2630X + 10,600	0.9997	6.69–5390	1.26	4.20	4.65	4.42	3.18	4.37	98.14	1.68	0.95
25	Leucine	Y = 3010X − 31,600	0.9991	13.30–6072	3.00	9.99	4.74	4.58	4.96	2.65	97.75	2.01	0.98
26	Thymidine	Y = 1390X − 23,200	0.9990	19.10–5070	3.78	12.60	4.58	4.81	4.66	4.83	100.29	1.78	1.03
27	Phenylalanine	Y = 1380X + 30,600	0.9999	5.56–34,920	1.62	5.40	1.30	1.14	3.45	1.45	99.34	0.12	1.04
28	Protocatechuic acid	Y = 2640X − 38,300	0.9990	15.40–5180	3.03	10.10	0.96	0.88	1.96	3.48	97.74	4.09	0.93
29	Neochlorogenic acid	Y = 617X + 1810	0.9997	2.80–4800	0.16	0.52	4.74	4.63	3.60	4.89	96.14	3.50	0.96
30	Procyanidin B2	Y = 21X − 1360	0.9997	82.30–54,800	13.59	45.30	4.96	4.97	3.47	0.89	99.96	1.09	0.99
31	3,4-Dihydroxybenzaldehyde	Y = 2420X + 9420	0.9999	0.50–5770	0.12	0.40	3.11	2.78	4.47	4.80	97.56	2.57	0.97
32	Epigallocatechin	Y = 373X − 503	0.9999	2.73–6204	0.58	1.92	4.43	4.94	3.71	2.42	97.49	3.80	1.01
33	Catechin	Y = 383X − 9360	0.9997	37–52,200	0.66	2.20	3.07	2.95	3.32	4.99	96.95	2.75	1.01
34	Procyanidin B1	Y = 3190X − 41,900	0.9992	14.90–5375	0.22	0.72	3.83	4.21	4.84	3.77	98.64	1.21	1.03
35	Chlorogenic acid	Y = 3240X − 3950	0.9997	3.03–4920	0.38	1.26	3.20	3.29	4.84	4.23	96.94	1.41	0.95
36	Cryptochlorogenic acid	Y = 422X + 9830	0.9999	1.44–19,860	0.35	1.16	2.32	2.50	3.75	4.29	97.92	3.38	0.92
37	Caffeic acid	Y = 3070X − 39,100	0.9996	15.10–5260	2.27	7.56	2.90	2.67	4.71	4.02	97.06	2.23	0.97
38	Epicatechin	Y = 488X − 14,600	0.9996	39.80–41,360	0.63	2.10	3.73	3.87	4.90	3.58	98.56	2.84	1.05
39	Polydatin	Y = 1600X − 17,600	0.9999	12–5310	3.18	10.60	3.17	3.03	3.09	4.93	101.00	4.62	0.96
40	Orientin	Y = 521X − 1260	0.9996	8.30–5170	0.10	0.34	4.54	4.40	4.91	4.95	100.33	2.68	1.01
41	Isoorientin	Y = 929X + 7750	0.9996	5.36–5840	0.12	0.39	4.54	4.38	4.60	2.02	100.19	3.10	0.93
42	Piceatannol	Y = 1110X − 20,600	0.9999	21.50–29,880	2.12	7.07	3.00	3.35	1.68	4.14	100.40	4.11	1.04
43	Vitexin	Y = 2850X + 2430	0.9996	2.32–5590	0.08	0.28	2.86	2.84	1.08	4.64	100.94	3.05	0.97
44	Vitexin-2″-O-rhamnoside	Y = 789X − 1580	0.9991	4.80–5280	0.64	2.14	3.53	3.64	4.26	4.62	100.50	1.73	0.95
45	Isovitexin	Y = 2090X + 7750	0.9994	1.68–4970	0.11	0.36	3.28	2.67	4.32	4.47	101.39	4.41	0.92
46	Hyperoside	Y = 2540X + 46,700	0.9997	4.25–5616	0.30	1.01	4.97	4.79	3.06	4.70	100.08	3.41	1.03
47	Aromadendrin	Y = 941X − 2560	0.9995	4.60–4850	0.41	1.36	4.97	4.86	3.38	4.93	97.84	3.84	1.02
48	Rutin	Y = 701X + 24,700	0.9995	21.50–51,000	1.24	4.12	1.36	1.46	2.42	4.96	100.18	2.53	0.94
49	Isoquercitrin	Y = 1720X + 246,000	0.9994	25.50–21,880	0.46	1.53	1.50	1.26	1.00	4.69	100.02	2.01	0.97
50	Resveratrol	Y = 93.2X + 184	0.9996	1.18–8288	0.23	0.78	4.78	4.48	4.93	4.75	100.53	4.87	0.98
51	Quercitrin	Y = 2500X − 10,500	0.9993	10.10–5160	1.60	5.33	2.09	1.98	4.54	2.91	98.49	1.35	1.05
52	Astragalin	Y = 1380X + 9790	0.9991	16.50–30,120	0.03	0.09	2.46	2.13	3.36	4.65	98.44	3.38	0.93
53	Nicotifiorin	Y = 806X + 2030	0.9994	8.34–29,760	1.07	3.58	2.46	2.13	1.58	4.82	96.85	4.76	0.96
54	Narcissin	Y = 952X − 19,100	0.9994	23.90–4910	0.69	2.31	2.04	1.75	3.02	4.82	98.16	3.10	0.95
55	Afzelin	Y = 1140X + 3300	0.9996	6.95–8660	0.08	0.27	2.81	2.67	3.73	4.16	98.66	3.77	0.92
56	Quercetin	Y = 3280X − 5780	0.9995	3.72–5160	0.77	2.56	4.24	4.31	2.26	4.68	98.37	3.52	0.99
57	Luteolin	Y = 5040X − 24,500	0.9992	6.47–5640	1.23	4.10	3.74	3.78	4.60	4.99	98.49	1.35	1.04
58	Kaempferol	Y = 280X − 3160	0.9997	13.50–2560	2.48	8.26	4.47	4.65	4.75	3.14	100.17	1.31	1.03
59	Apigenin	Y = 3130X − 342	0.9999	1.28–5320	0.09	0.29	3.77	3.44	2.96	3.03	98.29	2.28	0.98
60	Isorhamnetin	Y = 2330X − 4310	0.9996	4.37–5310	0.57	1.89	2.56	2.61	4.77	4.61	98.23	1.37	0.96

**Table 3 molecules-27-04813-t003:** Quality sequencing of the 47 tested samples.

No.	*r_i_*	Ranking	Difference of (*r_i_*%)	No.	*r_i_*	Ranking	Difference of (*r_i_*%)
S1	0.3363	38	30.80	S24	0.3621	31	25.47
S2	0.3925	9	19.23	S25	0.3485	35	28.29
S3	0.3743	21	22.97	S26	0.3610	32	25.71
S4	0.4471	4	7.98	S27	0.3480	36	28.37
S5	0.4499	3	7.41	S28	0.3638	28	25.13
S6	0.3812	16	21.56	S29	0.3655	27	24.79
S7	0.4007	6	17.55	S30	0.3146	43	35.26
S8	0.3919	12	19.35	S31	0.2980	47	38.67
S9	0.3858	14	20.60	S32	0.3771	17	22.40
S10	0.3673	24	24.40	S33	0.3656	26	24.75
S11	0.3716	22	23.53	S34	0.3921	11	19.31
S12	0.3660	25	24.68	S35	0.3635	29	25.20
S13	0.3621	30	25.47	S36	0.3224	41	33.65
S14	0.3923	10	19.26	S37	0.3105	44	36.09
S15	0.3506	33	27.84	S38	0.3097	45	36.26
S16	0.3974	8	18.22	S39	0.3049	46	37.25
S17	0.4859	1	0.00	S40	0.3676	23	24.35
S18	0.3764	20	22.54	S41	0.3176	42	34.63
S19	0.4024	5	17.19	S42	0.3304	39	32.00
S20	0.3844	15	20.90	S43	0.3765	19	22.51
S21	0.3887	13	20.01	S44	0.3441	37	29.18
S22	0.4580	2	5.75	S45	0.3284	40	32.41
S23	0.3997	7	17.73	S46	0.3493	34	28.11
				S47	0.3767	18	22.47

**Table 4 molecules-27-04813-t004:** Information of ES and WS.

Samples	No.	Habitats	Samples	No.	Habitats
ES	S1	Fuzhou, Fujian	WS	S24	Shizhu, Chongqing
	S2	Sanming, Fujian		S25	Shizhu, Chongqing
	S3	Sanming, Fujian		S26	Shizhu, Chongqing
	S4	Sanming, Fujian		S27	Guilin, Guangxi
	S5	Quanzhou, Fujian		S28	Guilin, Guangxi
	S6	Quanzhou, Fujian		S29	Guilin, Guangxi
	S7	Yongzhou, Hunan		S30	Guilin, Guangxi
	S8	Yongzhou, Hunan		S31	Baise, Guangxi
	S9	Ganzhou, Jiangxi		S32	Baise, Guangxi
	S10	Ganzhou, Jiangxi		S33	Baise, Guangxi
	S11	Ganzhou, Jiangxi		S34	Baise, Guangxi
	S12	Ganzhou, Jiangxi		S35	Baise, Guangxi
	S13	Ganzhou, Jiangxi		S36	Luodian, Guizhou
	S14	Taizhou, Zhejiang		S37	Luodian, Guizhou
	S15	Taizhou, Zhejiang		S38	Luodian, Guizhou
	S16	Ningbo, Zhejiang		S39	Luodian, Guizhou
	S17	Ningbo, Zhejiang		S40	Luodian, Guizhou
	S18	Ningbo, Zhejiang		S41	Luodian, Guizhou
	S19	Ningbo, Zhejiang		S42	Xingren, Guizhou
	S20	Lishui, Zhejiang		S43	Xingren, Guizhou
	S21	Lishui, Zhejiang		S44	Enshi, Hubei
	S22	Lishui, Zhejiang		S45	Guangnan, Yunnan
	S23	Zhoushan, Zhejiang		S46	Guangnan, Yunnan
				S47	Guangnan, Yunnan

## Data Availability

The data presented in this study are available in Appendix A.

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
