# Peer review of "Quality Evaluation of Tetrastigmae Radix from Two Different Habitats Based on Simultaneous Determination of Multiple Bioactive Constituents Combined with Multivariate Statistical Analysis"

_molecules, 2022, doi:10.3390/molecules27154813_

Round 1

Reviewer 1 Report

The authors developed a method to simultaneously annotate the phytochemical constituents of an important medicinal plant that is widely utilized in traditional Chinese medicine. However, some points need to be improved.

1. The authors should highlight the differences between their method and the previously used method.

2. The authors should highlight the limitation of the developed method and the effect of the extraction (60% ethanol) on the diverse metabolites, especially the very polar ones.

3. The spectra of the authentic compounds should be provided as a supplementary file in both negative and positive modes.

Reviewer 2 Report

The authors of the submitted manuscript claim that the content of active substances in plants from different habitats is different. The developed and presented methodology confirms this thesis and enables the analysis of individual samples in terms of their quality. The work is interesting, however, only researchers in the field of Chinese medicine product analysis will be interested in.

Comments:

Was it based on the existing literature while optimizing the conditions of the analysis, or did you make your own attempts to determine what phases (stationary, mobile, etc.) and other analysis conditions to use?

chapter 4.3. - As all concentrations were determined to be close to 1 mg / ml, such information could be given here, and the individual concentration values should be reported as Supplementary Data.

According to what guidelines was the method validated?

The legend should be completed in the Tables (defining the used symbols).

Reviewer 3 Report

The article fits into the Journal's scope. Subject matter is important. The topic of the research is interesting and worth of attention. All research components are present and clearly stated. The abstract is concise and clear. Introduction is well‐written and accurate. Objective of the study is clearly presented. Conclusions are drawn from the analysis of the collected data. References are adequate and based on relevant literature.

This article is a piece of a really, really good job.

Round 2

Reviewer 1 Report

The authors have not provided the spectra of the authentic compounds, please provide (the MS2 of each individual compound, not the chromatogram but the fragmentation pattern)
